# Vision Transformers Need Registers

**Timothée Darcet**[1,2]        **Maxime Oquab**[1]        **Julien Mairal**[2]        **Piotr Bojanowski**[1]

[1] FAIR, Meta
[2] Univ. Grenoble Alpes, Inria, CNRS, Grenoble INP, LJK, 38000 Grenoble, France

## Abstract

Transformers have recently emerged as a powerful tool for learning visual representations. In this paper, we identify and characterize artifacts in feature maps of both supervised and self-supervised ViT networks. The artifacts correspond to high-norm tokens appearing during inference primarily in low-informative background areas of images, that are repurposed for internal computations. We propose a simple yet effective solution based on providing additional tokens to the input sequence of the Vision Transformer to fill that role. We show that this solution fixes that problem entirely for both supervised and self-supervised models, sets a new state of the art for self-supervised visual models on dense visual prediction tasks, enables object discovery methods with larger models, and most importantly leads to smoother feature maps and attention maps for downstream visual processing.

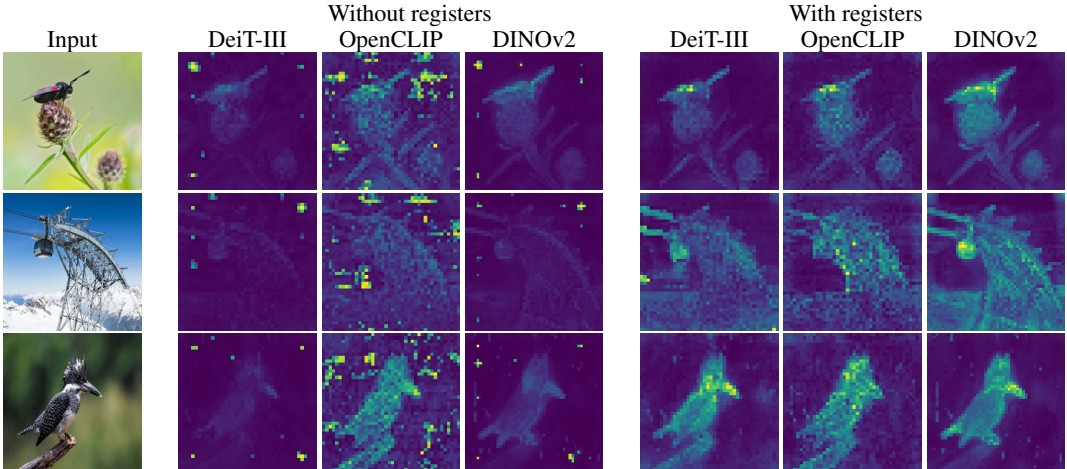

Figure 1: Register tokens enable interpretable attention maps in all vision transformers, similar to the original DINO method (Caron et al., 2021). Attention maps are calculated in high resolution for better visualisation. More qualitative results are available in appendix H.

## 1 Introduction

Embedding images into generic features that can serve multiple purposes in computer vision has been a long-standing problem. First methods relied on handcrafted principles, such as SIFT (Lowe, 2004), before the scale of data and deep learning techniques allowed for end-to-end training. Pursuing generic feature embeddings is still relevant today, as collecting valuable annotated data for many specific tasks remains difficult. This difficulty arises because of the required expertise (*e.g.*, medical data, or remote sensing) or the cost at scale. Today, it is common to pretrain a model for a task for which plenty of data is available and extract a subset of the model to use as a feature extractor. Multiple approaches offer this possibility; supervised methods, building on classification

Correspondence to `timdarcet@meta.com`

| Input | DeiT-III-B | DeiT-III-L | OpenCLIP-B | OpenCLIP-L | DINO-B | DINOv2-g |
|-------|-----------|-----------|-----------|-----------|--------|----------|

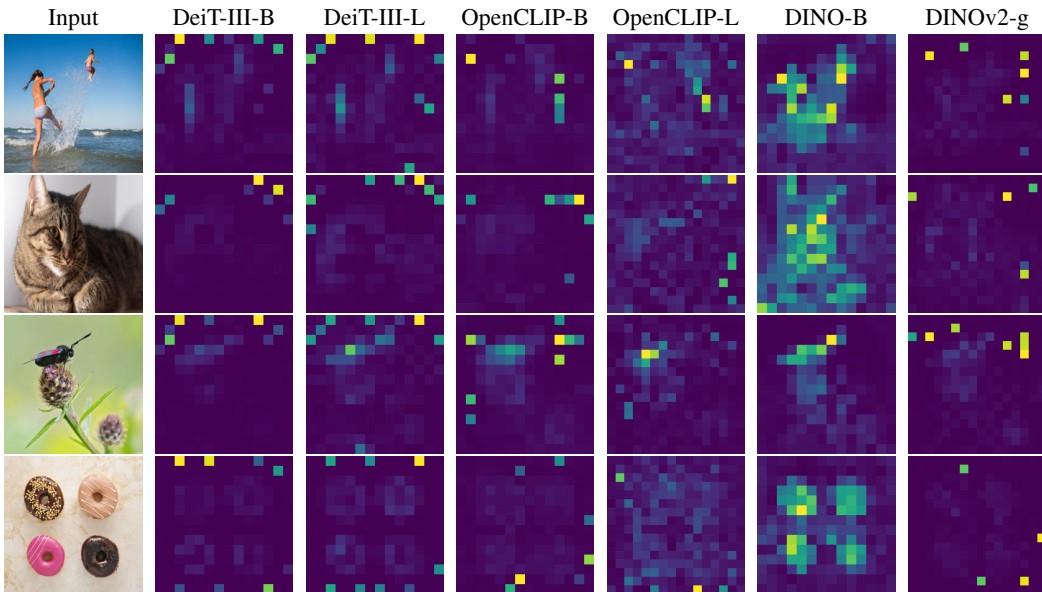

Figure 2: Illustration of artifacts observed in the attention maps of modern vision transformers. We consider ViTs trained with label supervision (DeiT-III), text-supervision (OpenCLIP) or self-supervision (DINO and DINOv2). Interestingly, all models but DINO exhibit peaky outlier values in the attention maps. The goal of this work is to understand and mitigate this phenomenon.

or text-image alignment, allow training strong feature models to unlock downstream tasks. Alternatively, self-supervised methods building on the Transformer architecture have attracted significant attention due to their high prediction performance on downstream tasks and the intriguing ability of some models to provide unsupervised segmentations (Caron et al., 2021)

In particular, the DINO algorithm is shown to produce models that contain explicit information about the semantic layout of an image. Indeed, qualitative results show that the last attention layer naturally focuses on semantically consistent parts of images and often produces interpretable attention maps. Exploiting these properties, object discovery algorithms such as LOST (Siméoni et al., 2021) build on top of DINO. Such algorithms can detect objects without supervision by gathering information in attention maps. They are effectively unlocking a new frontier in computer vision.

DINOv2 (Oquab et al., 2023), a follow-up to DINO, provides features that allow tackling dense prediction tasks. DINOv2 features lead to successful monocular depth estimation and semantic segmentation with a frozen backbone and linear models. Despite the strong performance on dense tasks, we observed that DINOv2 is surprisingly incompatible with LOST. When used to extract features, it delivers disappointing performance, only on par with supervised alternative backbones in this scenario. This suggests that DINOv2 behaves differently than DINO. The investigation described in this work notably exposes the presence of artefacts in the feature maps of DINOv2 that were not present in the first version of this model. These are observable qualitatively using straightforward methods. Also surprisingly, applying the same observations to supervised vision transformers exposes similar artifacts, as shown in Fig. 2. This suggests that DINO is, in fact, an exception, while DINOv2 models match the baseline behavior of vision transformers.

In this work, we set out to better understand this phenomenon and develop methods to detect these artifacts. We observe that they are tokens with roughly 10x higher norm at the output and correspond to a small fraction of the total sequence (around 2%). We also show that these tokens appear around the middle layers of the vision transformer, and that they only appear after a sufficiently long training of a sufficiently big transformer. In particular, we show that these outlier tokens appear in patches similar to their neighbors, meaning patches that convey little additional information.

As part of our investigation, we evaluate the outlier tokens with simple linear models to understand the information they contain. We observe that, compared to non-outlier tokens, they hold less information about their original position in the image or the original pixels in their patch. This ob-

Figure 3: Comparison of local feature norms for DINO ViT-B/16 and DINOv2 ViT-g/14. We observe that DINOv2 has a few outlier patches, whereas DINO does not present these artifacts. For DINOv2, although most patch tokens have a norm between 0 and 100, a small proportion of tokens have a very high norm. We measure the proportion of tokens with norm larger than 150 at 2.37%.

servation suggests that the model discards the local information contained in these patches during inference. On the other hand, learning an image classifier on outlier patches yields significantly stronger accuracy than doing so on the other patches, suggesting that they contain global information about the image. We propose the following interpretation to these elements: the model learns to recognize patches containing little useful information, and recycle the corresponding tokens to aggregate global image information while discarding spatial information.

This interpretation is consistent with an inner mechanism in transformer models that allows performing computations within a restricted set of tokens. In order to test this hypothesis, we append additional tokens - that we call registers - to the token sequence, independent of the input image. We train several models with and without this modification and observe that the outlier tokens disappear from the sequence entirely. As a result, the performance of the models increases in dense prediction tasks, and the resulting feature maps are significantly smoother. These smooth feature maps enable object discovery methods like LOST mentioned above with the updated models.

## 2 PROBLEM FORMULATION

As shown in Fig. 2, most modern vision transformers exhibit artifacts in the attention maps. The unsupervised DINO backbone (Caron et al., 2021) has been previously praised for the quality of local features and interpretability of attention maps. Surprisingly, the outputs of the subsequent DINOv2 models have been shown to hold good local information but exhibit undesirable artifacts in attention maps. In this section, we propose to study *why* and *when* these artifacts appear. While this work focuses on alleviating artefacts in all vision transformers, we focus our analysis on DINOv2.

### 2.1 ARTIFACTS IN THE LOCAL FEATURES OF DINOv2

**Artifacts are high-norm outlier tokens.** We want to find a quantitative way of characterizing artefacts that appear in the local features. We observe that an important difference between "artifact" patches and other patches is the norm of their token embedding at the output of the model. In Fig. 3 (left), we compare the norm of local features for a DINO and DINOv2 model given a reference image. We clearly see that the norm of artifact patches is much higher than the norm of other patches. We also plot the distribution of feature norms over a small dataset of images in Fig. 3 (right), which is clearly bimodal, allowing us to choose a simple criterion for the rest of this section: tokens with norm higher than 150 will be considered as "high-norm" tokens, and we will study their properties relative to regular tokens. This hand-picked cutoff value can vary across models. In the rest of this work, we use "high-norm" and "outlier" interchangeably.

**Outliers appear during the training of large models.** We make several additional observations about the conditions in which these outlier patches appear during the training of DINOv2. This analysis is illustrated in Fig. 4. First, these high-norm patches seem to differentiate themselves from other patches around layer 15 of this 40-layer ViT (Fig. 4a). Second, when looking at the distribution of norms along training of DINOv2, we see that these outliers only appear after one third of training (Fig. 4b). Finally, when analyzing more closely models of different size (Tiny, Small, Base, Large, Huge and giant), we see that only the three largest models exhibit outliers (Fig. 4c).

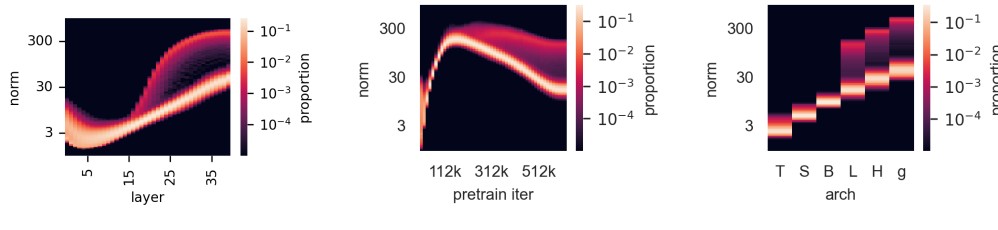

(a) Norms along layers.  (b) Norms along iterations.  (c) Norms across model size.

Figure 4: Illustration of several properties of outlier tokens in the 40-layer DINOv2 ViT-g model. **(a)**: Distribution of output token norms along layers. **(b)**: Distribution of norms along training iterations. **(c)**: Distribution of norms for different model sizes. The outliers appear around the middle of the model during training; they appear with models larger than and including ViT-Large.

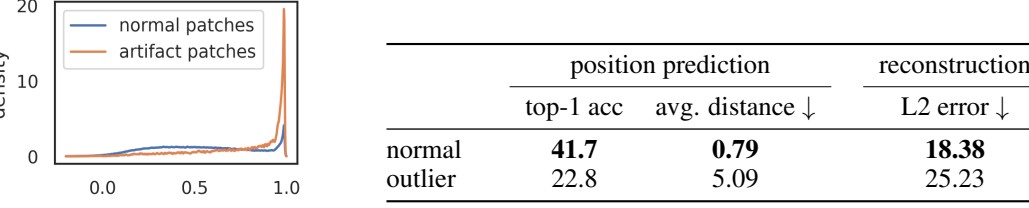

| | position prediction | | reconstruction |
|---|---|---|---|
| | top-1 acc | avg. distance ↓ | L2 error ↓ |
| normal | **41.7** | **0.79** | **18.38** |
| outlier | 22.8 | 5.09 | 25.23 |

(a) Cosine similarity to neighbors.    (b) Linear probing for local information.

Figure 5: **(a)**: Distribution of cosine similarity between input patches and their 4 neighbors. We plot separately artifact patches (norm of the *output token* over 150) and normal patches. **(b)**: Local information probing on normal and outlier patch tokens. We train two models: one for predicting position, and one for reconstructing the input patch. Outlier tokens have much lower scores than the other tokens, suggesting they are storing less local patch information.

**High-norm tokens appear where patch information is redundant.** To verify this, we measure the cosine similarity between high-norm tokens and their 4 neighbors right after the patch embedding layer (at the beginning of the vision transformer). We illustrate the density plot in Fig. 5a. We observe that high-norm tokens appear on patches that are very similar to their neighbors. This suggests that these patches contrain redundant information and that the model could discard their information without hurting the quality of the image representation. This matches qualitative observations (see Fig. 2) that they often appear in uniform, background areas.

**High-norm tokens hold little local information.** In order to better understand the nature of these tokens, we propose to probe the patch embeddings for different types of information. For that we consider two different tasks: position prediction and pixel reconstruction. For each of these tasks, we train a linear model on top of the patch embeddings, and measure the performance of this model. We compare the performance achieved with high-norm tokens and with other tokens, to see if high-norm tokens contain different information than "normal" tokens.

- **Position prediction.** We train a linear model to predict the position of each patch token in the image, and measure its accuracy. We note that this position information was injected in the tokens before the first ViT layer in the form of absolute position embeddings. We observe that high-norm tokens have much lower accuracy than the other tokens (Fig. 5b), suggesting they contain less information about their position in the image.

- **Pixel reconstruction.** We train a linear model to predict the pixel values of the image from the patch embeddings, and measure the accuracy of this model. We observe again that high-norm tokens achieve much lower accuracy than other tokens (Fig. 5b). This suggests that high-norm tokens contain less information to reconstruct the image than the others.

**Artifacts hold global information.** In order to evaluate how much global information is gathered in the high-norm tokens, we propose to evaluate them on standard image representation learning

|  | IN1k | P205 | Airc. | CF10 | CF100 | CUB | Cal101 | Cars | DTD | Flow. | Food | Pets | SUN | VOC |
|---|---|---|---|---|---|---|---|---|---|---|---|---|---|---|
| [CLS] | **86.0** | **66.4** | **87.3** | **99.4** | **94.5** | **91.3** | 96.9 | **91.5** | **85.2** | **99.7** | **94.7** | **96.9** | **78.6** | 89.1 |
| normal | 65.8 | 53.1 | 17.1 | 97.1 | 81.3 | 18.6 | 73.2 | 10.8 | 63.1 | 59.5 | 74.2 | 47.8 | 37.7 | 70.8 |
| outlier | 69.0 | 55.1 | 79.1 | 99.3 | 93.7 | 84.9 | **97.6** | 85.2 | 84.9 | 99.6 | 93.5 | 94.1 | 78.5 | **89.7** |

Table 1: Image classification via linear probing on normal and outlier patch tokens. We also report the accuracy of classifiers learnt on the class token. We see that outlier tokens have a much higher accuracy than regular ones, suggesting they are effectively storing global image information.

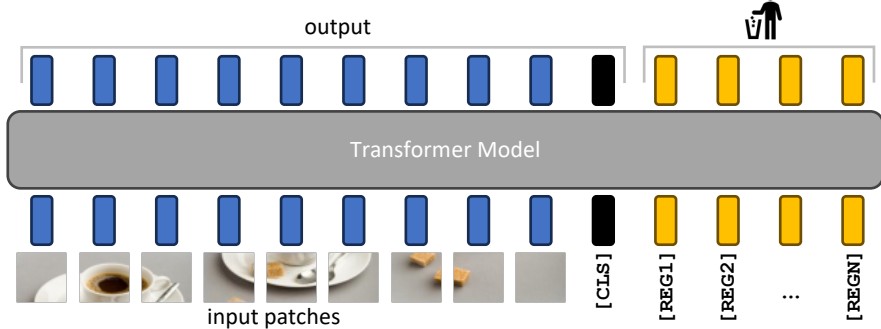

Figure 6: Illustration of the proposed remediation and resulting model. We add $N$ additional learnable input tokens (depicted in yellow), that the model can use as *registers*. At the output of the model, only the patch tokens and [CLS] tokens are used, both during training and inference.

benchmarks. For each image in a classification dataset, we forward it through DINOv2-g and extract the patch embeddings. From those, we choose a single token at random, either high-norm or normal. This token is then considered as the image representation. We then train a logistic regression classifier to predict the image class from this representation, and measure the accuracy. We observe that the high-norm tokens have a much higher accuracy than the other tokens (Table 1). This suggests that outlier tokens contain more global information than other patch tokens.

## 2.2 HYPOTHESIS AND REMEDIATION

Having made these observations, we make the following hypothesis: *large*, *sufficiently trained* models learn to recognize *redundant* tokens, and to use them as places to *store*, *process* and *retrieve* global information. Furthermore, we posit that while this behavior is not bad in itself, the fact that it happens inside the patch tokens is undesirable. Indeed, it leads the model to discard local patch information (Tab. 5b), possibly incurring decreased performance on dense prediction tasks.

We therefore propose a simple fix to this issue: we explicitly add new tokens to the sequence, that the model can learn to use as registers. We add these tokens after the patch embedding layer, with a learnable value, similarly to the [CLS] token. At the end of the vision transformer, these tokens are discarded, and the [CLS] token and patch tokens are used as image representations, as usual. This mechanism was first proposed in Memory Transformers (Burtsev et al., 2020), improving translation tasks in NLP. Interestingly, we show here that this mechanism admits a natural justification for vision transformers, fixing an interpretability and performance issue that was present otherwise.

We note that we have not been able to fully determine which aspects of the training led to the appearance of artifacts in different models. The pretraining paradigm seems to play a role, as OpenCLIP and DeiT-III exhibit outliers both at size B and L (Fig. 2). However, the model size and training length also play important parts, as observed in Fig. 4.

## 3 EXPERIMENTS

In this section, we validate the proposed solution by training vision transformers with additional [reg] register tokens. We evaluate the effectiveness of our approach by a quantitative and qualitative analysis. We then ablate the number of registers used for training, to check that they do not

Figure 7: Effect of register tokens on the distribution of output norms on DINOv2, OpenCLIP and DeiT-III. Using register tokens effectively removes the norm outliers that were present previously.

cause a performance regression, evaluate an unsupervised object discovery method atop our features and finally provide a qualitative analysis of the patterns learnt by the registers.

## 3.1 TRAINING ALGORITHMS AND DATA

As the proposed solution is a simple architectural change, we can easily apply it to any training procedure. We try it on three different state-of-the-art training methods for supervised, text-supervised, and unsupervised learning, shortly described below.

**DEIT-III** (Touvron et al., 2022) is a simple and robust supervised training recipe for classification with ViTs on ImageNet-1k and ImageNet-22k. We choose this method as an example of label-supervised training as it is simple, uses the base ViT architecture, achieves strong classification results, and is easy to reproduce and modify with our improvements. We run this method on the ImageNet-22k dataset, using the ViT-B settings, as provided in the official repository [1].

**OpenCLIP** (Ilharco et al., 2021) is a strong training method for producing text-image aligned models, following the original CLIP work. We chose this method as an example of text-supervised training because it is open-source, uses the base ViT architecture, and is easy to reproduce and modify with our improvements. We run the OpenCLIP method on a text-image-aligned corpus based on Shutterstock that includes only licensed image and text data. We use a ViT-B/16 image encoder, as proposed in the official repository [2].

**DINOv2** (Oquab et al., 2023) is a self-supervised method for learning visual features, following the DINO work. We apply our changes to this method as it is the main focus of our study. We run this method on ImageNet-22k with the ViT-L configuration. We use the official repository [3].

## 3.2 EVALUATION OF THE PROPOSED SOLUTION

As shown in Fig. 1, we get rid of the artifacts by training models with additional register tokens. In the appendix, we provide additional qualitative results for more images in Fig. 19. In order to quantitatively measure this effect, for each model, we probe the norm of features at the output of the model. We report these norms for all three algorithms with and without registers in Fig. 7. We see that when training with registers, models do not exhibit large-norm tokens at the output, which confirms the initial qualitative assessment.

**Performance regression.** In the previous section, we have shown that the proposed approach removes artifacts from local feature maps. In this experiment, we want to check that the use of register tokens does not affect the representation quality of those features. We run linear probing on ImageNet classification, ADE20k Segmentation, and NYUd monocular depth estimation. We follow the experimental protocol outlined in Oquab et al. (2023). We summarize the performance of the models described in Sec. 3.1 with and without register tokens in Table 2a. We see that when using registers, models do not lose performance and sometimes even work better. For completeness, we also provided the zero-shot classification performance on ImageNet for OpenCLIP (Table 2b), which remains unchanged. Please note that the absolute performance of our OpenCLIP reproduction is lower due to the data source we used.

[1] https://github.com/facebookresearch/deit
[2] https://github.com/mlfoundations/open_clip
[3] https://github.com/facebookresearch/dinov2

|              | ImageNet Top-1 | ADE20k mIoU | NYUd rmse ↓ |
| ------------ | -------------- | ----------- | ----------- |
| DeiT-III     | 84.7           | 38.9        | 0.511       |
| DeiT-III+reg | 84.7           | 39.1        | 0.512       |
| OpenCLIP     | 78.2           | 26.6        | 0.702       |
| OpenCLIP+reg | 78.1           | 26.7        | 0.661       |
| DINOv2       | 84.3           | 46.6        | 0.378       |
| DINOv2+reg   | 84.8           | 47.9        | 0.366       |

(a) Linear evaluation with frozen features.

|              | ImageNet Top-1 |
| ------------ | -------------- |
| OpenCLIP     | 59.9           |
| OpenCLIP+reg | 60.1           |

(b) Zero-shot classification.

Table 2: Evaluation of downstream performance of the models that we trained, with and without registers. We consider linear probing of frozen features for all three models, and zero-shot evaluation for the OpenCLIP model. We see that using register not only does not degrade performance, but even improves it by a slight margin in some cases.

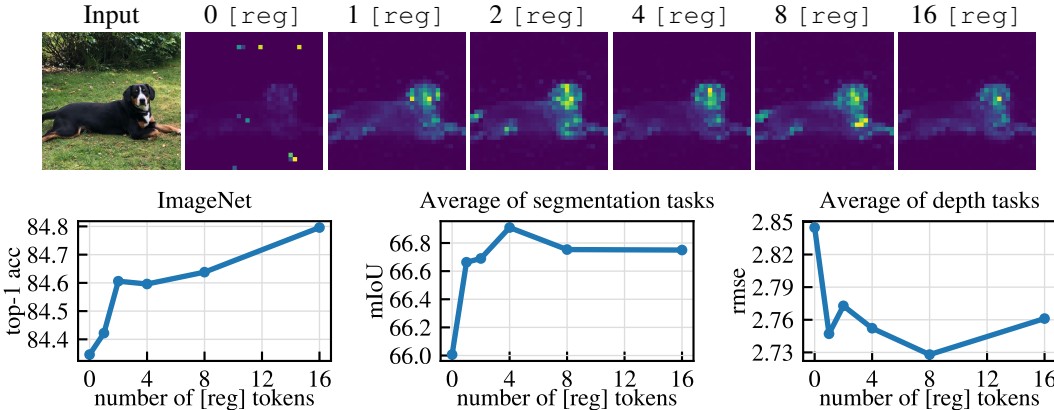

Figure 8: Ablation of the the number of register tokens used with a DINOv2 model. **(top)**: qualitative visualization of artifacts appearing as a function of number of registers. **(bottom)**: performance on three tasks (ImageNet, ADE-20k and NYUd) as a function of number of registers used. While one register is sufficient to remove artefacts, using more leads to improved downstream performance.

**Number of register tokens.** As described in Sec. 2.2, we propose alleviating the feature maps' artifacts by adding register tokens. In this experiment, we study the influence of the number of such tokens on local features and downstream performance. We train DINOv2 ViT-L/14 models with 0, 1, 2, 4, 8 or 16 registers. In Fig. 8, we report the results of this analysis. In Fig. 8**(top)**, we qualitatively study the attention maps and observe that the visible artifacts disappear when adding at least one register. We then examine in Fig. 8**(bottom)** performance on downstream evaluation benchmarks, following the protocol from Oquab et al. (2023). There seems to be an optimal number of registers for dense tasks, and adding one brings most of the benefit. This optimum is likely explained by the disappearance of artifacts, leading to better local features. On ImageNet, however, performance improves when using more registers. In all our experiments, we kept 4 register tokens.

## 3.3 OBJECT DISCOVERY

Recent unsupervised object discovery methods rely on the quality and smoothness of local feature maps (Siméoni et al., 2021; Wang et al., 2023). By leveraging DINO Caron et al. (2021), these methods have significantly surpassed the previous state of the art. However, the algorithm leads to poor performance when applied to modern backbones such as DINOv2 Oquab et al. (2023) or supervised ones Touvron et al. (2022). We posit that this can be alleviated by the method proposed in this work. We run LOST (Siméoni et al., 2021) on features extracted from backbones trained using the algorithms described in Sec.3.1 with and without registers. We run object discovery on PASCAL

|  | VOC 2007 | VOC 2012 | COCO 20k |
|---|---|---|---|
| DeiT-III | 11.7 | 13.1 | 10.7 |
| DeiT-III+reg | 27.1 | 32.7 | 25.1 |
| OpenCLIP | 38.8 | 44.3 | 31.0 |
| OpenCLIP+reg | 37.1 | 42.0 | 27.9 |
| DINOv2 | 35.3 | 40.2 | 26.9 |
| DINOv2+reg | 55.4 | 60.0 | 42.0 |

Table 3: Unsupervised Object Discovery using LOST (Siméoni et al., 2021) on models with and without registers. We evaluated three types of models trained with various amounts of supervision on VOC 2007, 2012 and COCO. We measure performance using corloc. We observe that adding register tokens makes all models significantly more viable for usage in object discovery.

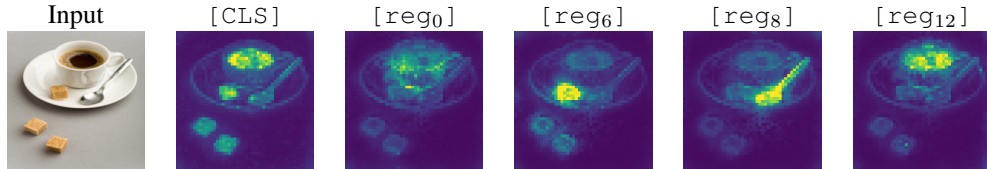

Figure 9: Comparison of the attention maps of the `[CLS]` and register tokens. Register tokens sometimes attend to different parts of the feature map, similarly to slot attention (Locatello et al., 2020). This behaviour was never required from the model, and emerged naturally from training.

VOC 2007 and 2012 and COCO 20k. We use values for DeiT and OpenCLIP, and for DINOv2, we use keys. Because the output features may have different conditioning, we manually add a bias to the gram matrix of features. The results of this experiment are presented in Table 3. For DINOv2 and DeiT-III, adding registers significantly improves the discovery performance. For OpenCLIP, the performance is slighty worse with registers (see Sec. C for analysis). The performance of DINOv2 on VOC2007 still does not match that of DINO as reported by Siméoni et al. (2021) (61.9 corloc). However, the model with registers gets an improvement of 20.1 corloc (55.4 versus 35.3).

### 3.4 QUALITATIVE EVALUATION OF REGISTERS

In this final experiment, we qualitatively probe for the behavior of register tokens. We want to verify if they all exhibit similar attention patterns or whether a differentiation automatically emerges. To this end, we plot the attention maps of the class and register tokens to patch tokens. The result of this visualization is shown in Fig. 9. We see that registers do not have a completely aligned behavior. Some selected registers exhibit interesting attention patterns, attending to the different objects in the scene. While nothing enforced this behavior, their activations had some natural diversity. We leave the study of the regularization of registers for future work.

## 4 RELATED WORK

**Feature extraction with pretrained models.** Using pretrained neural network models for extracting visual features has stood the test of time since the AlexNet (Krizhevsky et al., 2012) CNN model pretrained on ImageNet-1k (Russakovsky et al., 2015). More recent models have upgraded the same setup with modern architectures, such as ResNets (used in, *e.g.*, DETR, Carion et al., 2020) or even Vision Transformers. As Transformers are easily able to handle different modalities during training, off-the-shelf backbones are now commonly trained on label supervision (*e.g.*, DeiT-III on ImageNet-22k, Touvron et al., 2022) or text supervision (e.g., CLIP (Radford et al., 2021)), providing strong *visual foundation models*, scaling well with model sizes, and enabling excellent performance on a variety of tasks including detection (Carion et al., 2020) and segmentation (Zheng et al., 2021; Kirillov et al., 2023). In this context, supervision relies on annotations in the form of labels or text alignment; the dataset biases (Torralba & Efros, 2011) are not well characterized, yet they drive

learning and shape the learned models. An alternative approach consists of not using supervision and letting the models learn from the data via a pretext task that is designed to require understanding the content of images (Doersch et al., 2015). This self-supervised learning paradigm was explored in multiple methods using Vision Transformers: MAE (He et al., 2022) trains a model at reconstructing pixel values of hidden areas of an image and then applies fine-tuning to address a new task. With a different approach, the self-distillation family of methods (He et al., 2020; Caron et al., 2021; Zhou et al., 2022) showcase strong performance using frozen backbones, allowing for more robustness to domain shifts for task-specific downstream models. In this work, we focused the analysis on self-supervised learning, and more specifically on the DINOv2 approach (Oquab et al., 2023), which has shown to be particularly effective for learning local features. We showed that despite excellent benchmark scores, DINOv2 features exhibit undesirable artifacts and that correcting these artifacts in the learning process allows for further improvements in the benchmark performances. These phenomenon is even more surprising as DINOv2 builds upon DINO (Caron et al., 2021), which does not show signs of artifacts. We then further showed that the correction techniques hold for supervised paradigms by testing on DeiT-III and OpenCLIP.

**Additional tokens in transformers.** Extending the transformer sequence with special tokens was popularized in BERT (Devlin et al., 2019). However, most approaches add new tokens either to provide the network with new information as for example `[SEP]` tokens in BERT, provide opportunity to spend more computation on the input as seen with the tape tokens in AdaTape (Xue et al., 2023), or to gather information in these tokens, and use their output value as an output of the model: for classification, as `[CLS]` tokens in BERT and ViT (Dosovitskiy et al., 2021); for generative learning, as `[MASK]` in BERT and BEiT (Bao et al., 2021); for detection, as object queries in DETR (Carion et al., 2020), detection tokens in YOLOS (Fang et al., 2021), and ViDT (Song et al., 2021); or for accumulating information from possibly multiple modalities before decoding, as latent token arrays in Perceivers (Jaegle et al., 2021; 2022). Different to these works, the tokens we add to the sequence add no information, and their output value is not used for any purpose. They are simply registers where the model can learn to store and retrieve information during the forward pass. The Memory Transformer (Burtsev et al., 2020), closer to our work, presents a simple approach to improve transformer models using memory tokens added to the token sequence, improving translation performance. In follow-up work, Bulatov et al. (2022) address complex copy-repeat-reverse tasks. Sandler et al. (2022) extend this line to the vision domain for fine-tuning but observe that such tokens do not transfer well across tasks. In contrast, we do not perform fine-tuning and employ additional tokens during pretraining to improve the features obtained for all tasks downstream. More importantly, our study contributes the following new insight in Sec. 2: the mechanism implemented through memory tokens already appears naturally in Vision Transformers; our study shows that *such tokens allow us not to create but to isolate this existing behavior*, and thus avoid collateral side-effects.

**Attention maps of vision transformers.** Visualising the attention map from `[CLS]` token to patch tokens was popularized in DINO (Caron et al., 2021). It was shown there that the attention maps of DINO were clean of artifacts, as opposed to the attention maps of previous vision transformers. Other works have since reported interesting attention maps using various techniques: by modifying the optimisation procedure (Chen et al., 2022), by steering the attention scores towards useful image parts (Shi et al., 2023), by modifying the architecture of the transformer layers (Yu et al., 2024), or by introducing a learnable pooling to produce the `[CLS]` token (Psomas et al., 2023).

## 5 CONCLUSION

In this work, we exposed artifacts in the feature maps of DINOv2 models, and found this phenomenon to be present in multiple existing popular models. We have described a simple method to detect these artifacts by observing that they correspond to tokens with an outlier norm value at the output of the Transformer model. Studying their location, we have proposed an interpretation that models naturally recycle tokens from low-informative areas and repurpose them into a different role for inference. Following this interpretation, we have proposed a simple fix, consisting of appending additional tokens to the input sequence that are not used as outputs, and have found that this entirely removes the artifacts, improving the performance in dense prediction and object discovery. Moreover, we have shown that the proposed solution also removes the same artifacts present in supervised models such as DeiT-III and OpenCLIP, confirming the generality of our solution.

ACKNOWLEDGMENTS

We thank Hu Xu, Oriane Siméoni, Mido Assran and Armand Joulin for their insightful discussions and help during the course of this work. We thank Pyrrhus for posing for fig 8. Julien Mairal was supported by ANR 3IA MIAI@Grenoble Alpes (ANR-19-P3IA-0003) and by ERC grant number 101087696 (APHELEIA project).

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

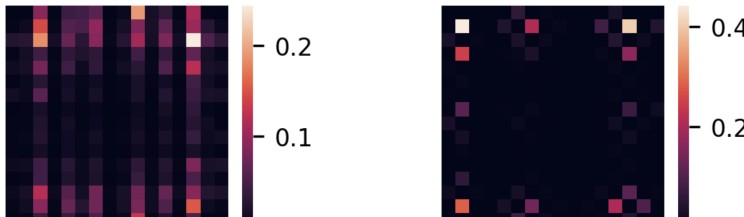

Figure 10: Feature norms along locations: proportion of tokens with norm larger than the cutoff value at a given location. Left: official DINOv2 model (no antialiasing), right: our models (with antialiasing). At some positions, more than 20% of tokens have a high norm.



Figure 11: Propagating unit gradients through a bicubic interpolation ($16 \times 16 \rightarrow 7 \times 7$) without antialiasing. We observe a striping pattern similar to the one of Fig. 10 (left).

## A    INTERPOLATION ARTIFACTS AND OUTLIER POSITION DISTRIBUTION

We plot in Figure 10 (left) the proportion of outlier tokens, characterized by a norm larger than the cutoff value defined manually, following the distribution of norms shown in Fig. 3 (main text). We make two observations:

First, the distribution has a vertical-striped pattern. We investigate this phenomenon and notice that in the original DINOv2 implementation, during training the position embeddings are interpolated from a $16 \times 16$ map into a $7 \times 7$ map, without antialiasing. Propagating unit gradients through such an interpolation function (bicubic resize) leads to the following gradients, shown in Fig. 11. In this work, when producing results with DINOv2 (especially for the results in Tables 2a,3), we always apply antialiasing in the interpolation operator, removing the striping pattern, which gives an updated distribution of outlier positions as shown in Fig. 10 (right).

Second, the outliers tend to appear in areas closer to the border of the feature map rather than in the center. Our interpretation is that the base model tends to recycle tokens in low-informative areas to use as registers: pictures produced by people tend to be object-centric, and in this case the border areas often correspond to background, which contains less information than the center.

## B    COMPLEXITY ANALYSIS

Since our proposed fix introduces new tokens, it also increases the number of learnable parameters and the FLOP count of the model. We show in Fig. 12 the relationship between number of registers and increase in model FLOP count and parameter count. We observe that adding registers induces a negligible change in number of parameters, and a slight change in FLOP count. Still, for $n = 4$ registers, the increase in FLOPs stays below 2%.

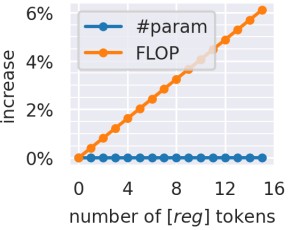

Figure 12: Increase in model parameter and FLOP count when adding different numbers of registers. Adding registers can increase model FLOP count by up to 6% for 16 registers. However, in the more common case of using 4 registers, that we use in most of our experiments, this increase is below 2%. In all cases, the increase in model parameters is negligible.

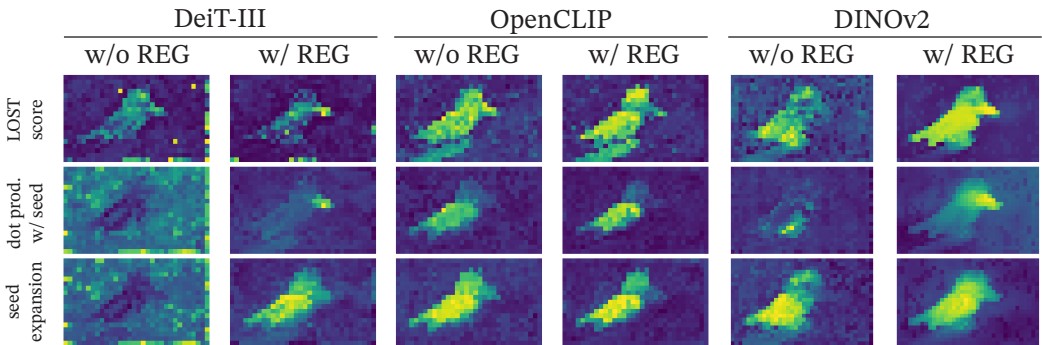

Figure 13: Illustration of the intermediate computations in the LOST algorithm for all models. Adding registers drastically improves the look of all intermediate steps for DeiT-III and DINOv2. The difference is less striking for the OpenCLIP model.

## C   ANALYSIS OF LOST PERFORMANCE

The results presented in Sec. 3.3 show that adding registers allows us to obtain better object discovery performance with DINOv2 models. The conclusions for the two other models studied in this work could be more crisp. In order to understand why this is so, we qualitatively study the impact of removing artifacts on the intermediate computations in the LOST algorithm. We show the intermediate outputs of LOST for all models on a given input image in Fig. 13.

Adding registers improves the scores and the resulting seed expansion for DeiT-III and DINOv2. This observation is coherent with the improved numbers reported in Table 3. For OpenCLIP, however, the LOST algorithm seems robust to the type of outliers observed in the local features. Adding registers does remove artifacts (as clearly shown in Fig. 20) but does not have much impact on the LOST score. It is also worth noting that OpenCLIP, with or without registers, provides comparable performance to DINOv2 without registers and DeiT-III with registers. The qualitative assessment is coherent with the numbers reported in Table 3.

A surprising observation is that despite the existence of high-norm patches in the output of Open-CLIP models without registers (as seen in Fig. 7), the seed expansion score in Fig. 13 looks smooth. In the LOST experiment with OpenCLIP models, we do not use the features directly, but the values from the computation of attention maps. In Fig. 14, we show the seed expansion score for Open-CLIP models with and without registers for keys, queries and values. We see that artifacts are clearly visible as spots in the background for keys and queries, for the model without registers. As soon as registers are used, the LOST score is focusing on the object, with a smoother score for values. We qualitatively observe that for the OpenCLIP model, the value projection filters out the outliers even without registers. This means that the outliers appear to live in the null space of the value projection layer; the investigation for this phenomenon is left for future work.

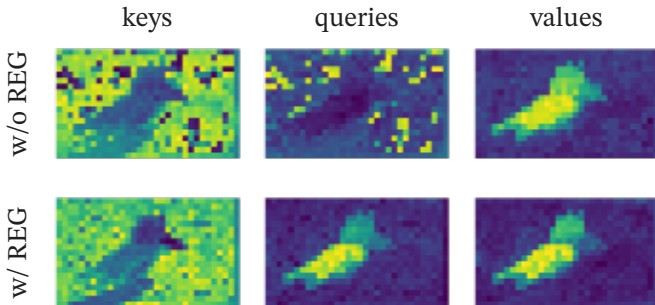

Figure 14: Illustration of the seed expansion score in LOST for an OpenCLIP model with and without registers for the three types of features considered: keys, queries, and values. The score is qualitatively improved across all features, with fewer artifacts appearing. Interestingly, the seed expansion map computed using values does not exhibit artifacts with nor without registers.

# D    BEHAVIOR OF MODELS TRAINED WITH REGISTERS

In order to better understand the phenomenon at hand, we examine the question of to what extent did the register tokens "replace" the high-norm tokens and took on the same role.

## D.1    NORMS

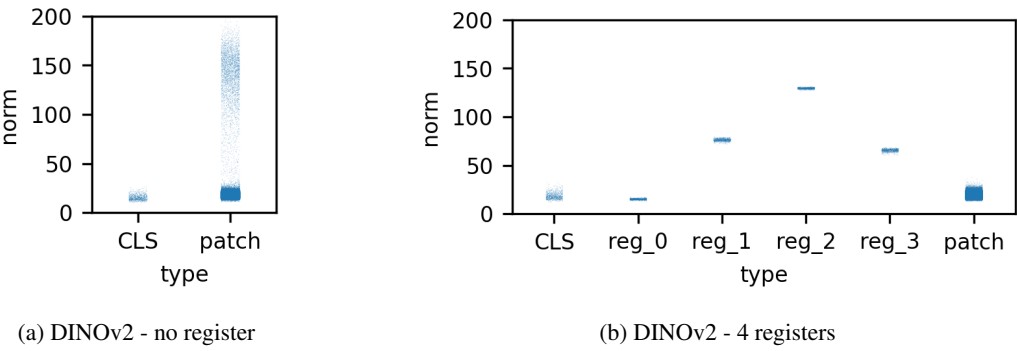

(a) DINOv2 - no register

(b) DINOv2 - 4 registers

Figure 15: Distribution of token norms for a DINOv2 model without (left) and with (right) 4 registers. Introducing registers entirely negates the high-norm outliers among the patch tokens.

In Fig. 15 we compare the distribution of token norms for a model with or without registers. This figure is similar to Fig. 7 but with a finer granularity, as we also plot the norm distribution of individual register tokens and `[CLS]` tokens. We observe the following: with registers, the norms of patch tokens do not contain outliers anymore, and the high-norm tokens are entirely contained in the set of registers. As a result, we conclude that the behavior leading to high-norm outliers in the model is effectively absorbed in the registers.

An additional interesting observation is that the norms of the registers appear to be quantized, compared to the previous outliers; we leave the investigation of this phenomenon for future work.

## D.2    INFORMATION HELD BY TOKENS

We report on table 4 the linear probing performance of models trained with and without registers, when using different tokens as representations. We evaluate on the aircrafts dataset, as it showed clear conclusions in the similar table 1. We observe that adding a register does not significantly modify the scores obtained with the `[CLS]` or patch tokens. However, the outlier patches are removed, and their behavior is transferred to the newly added register.

| #registers | [CLS] | top-1 accuracy | | |
|---|---|---|---|---|
| | | normal patch | outlier patch | register |
| 0 | 84.6 | 15.5 | 73.3 | N/A |
| 1 | 85.2 | 14.5 | N/A | 71.1 |

Table 4: Linear probing of models with and without registers on the Aircraft dataset, using various tokens as representation. We observe that the behavior of the outlier tokens, aggregating global information, is absorbed into the register.

We further conduct an evaluation of the local information contained in the patch tokens of a model trained with and without registers (table 5). We observe that the non-outliers patches, in both cases, hold similar local information, confirming that the registers only remove the outlier behavior, without significantly modifying the information held by the other patches.

| #registers | patches considered | position prediction top-1 acc | reconstruction L2 error $\downarrow$ |
|---|---|---|---|
| 0 | non-outliers | 66.3 | 15.9 |
| 4 | non-outliers (ie all) | 65.8 | 16.0 |

Table 5: Linear probing for local information on the patch tokens of models trained without or with registers. We only consider patches considered "normal", i.e. not the high-norm outliers. We observe that adding registers does not significantly modify the scores of these patches.

## D.3 POSITIONAL FOCUS

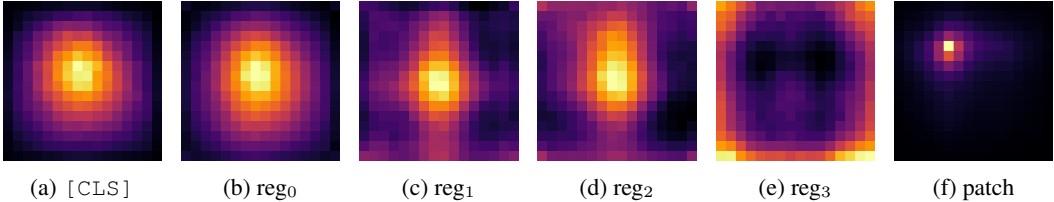

|    (a) [CLS]    |    (b) reg$_0$    |    (c) reg$_1$    |    (d) reg$_2$    |    (e) reg$_3$    |    (f) patch    |

Figure 16: Average attention map of registers and [CLS] token. There is a variability observed, with register 3 of this model focusing more on border areas. We also include the average attention map of a patch for comparison. The patch has a much more focused average attention.

In Fig. 16 we display the positional focus for the class token and the 4 registers of a DINOv2+reg model. We produce these plots by running the model on a random subset of ImageNet-22k, and averaging the attention maps for the corresponding tokens at the last layer. We note that ImageNet-22k contains mostly object-centric images rather than scenes, which explains why the average attention maps correspond to centered blobs.

We make several observations. First, the attention maps for registers can be different of each other; for example, register 3 tends to focus on border areas, while the other registers tend to focus on more centered areas. Register 2 tends to focus slightly more on the upper areas of images that others. This is consistent with Fig. 9, where we show registers focusing on different large areas of the image, suggesting some level of specialization.

Second, by comparing the register maps to the [CLS] token map and to a patch token map, we observe that registers produce maps with a large support area, very similarly to the [CLS] token, and very different of a typical patch token which is more localized. As the [CLS] token is known to carry global information (as proven by the linear probing classification performance): this suggests that registers also carry global information.

# E    MASKED AUTOENCODERS

Masked Autoencoding (He et al., 2022) is another common way of pretraining self-supervised models. We observe in Fig. 17 that there are no artifacts in the maps produced by MAE: our hypothesis is that the absence of artifacts is due to the training procedure using only a local loss on the patch tokens, rather than an objective involving global aggregation of information. However, we also note that the performance of MAE models is very low for self-supervised representation learning (75% linear probing performance on ImageNet classification for ViT-Large), preventing it from being used as is, and making fine-tuning a requirement.

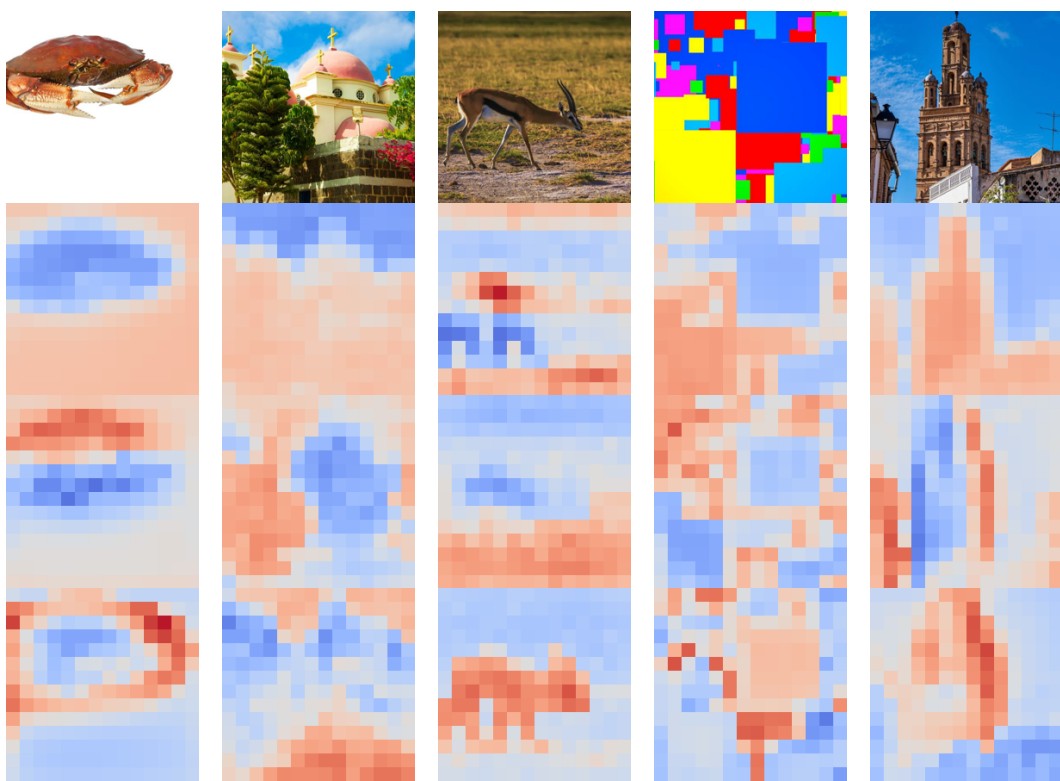

Figure 17: First three principal components of the output feature map of a ViT-Large Masked Autoencoder.

# F    BEHAVIOR PER ATTENTION HEAD

In this section, we investigate whether the artifacts appear only on the attention maps for specific heads of the last vision transformer block, or for all of them. We show in Fig. 18 the input image along with the attention maps for different heads. We observe that the artifacts appear for all attention heads, despite heads focusing on different areas of the object. We still observe that some heads focus more on artifacts than others.

# G    VARIANCE ON TOKEN INFORMATION PROBING

The results presented in table 1 are obtained by taking a random patch token, either normal or outlier. However, the choice of this token adds a significant source of variance in the evaluation. For thoroughness, we report in table 6 the standard deviation of the scores obtained relative to this choice.

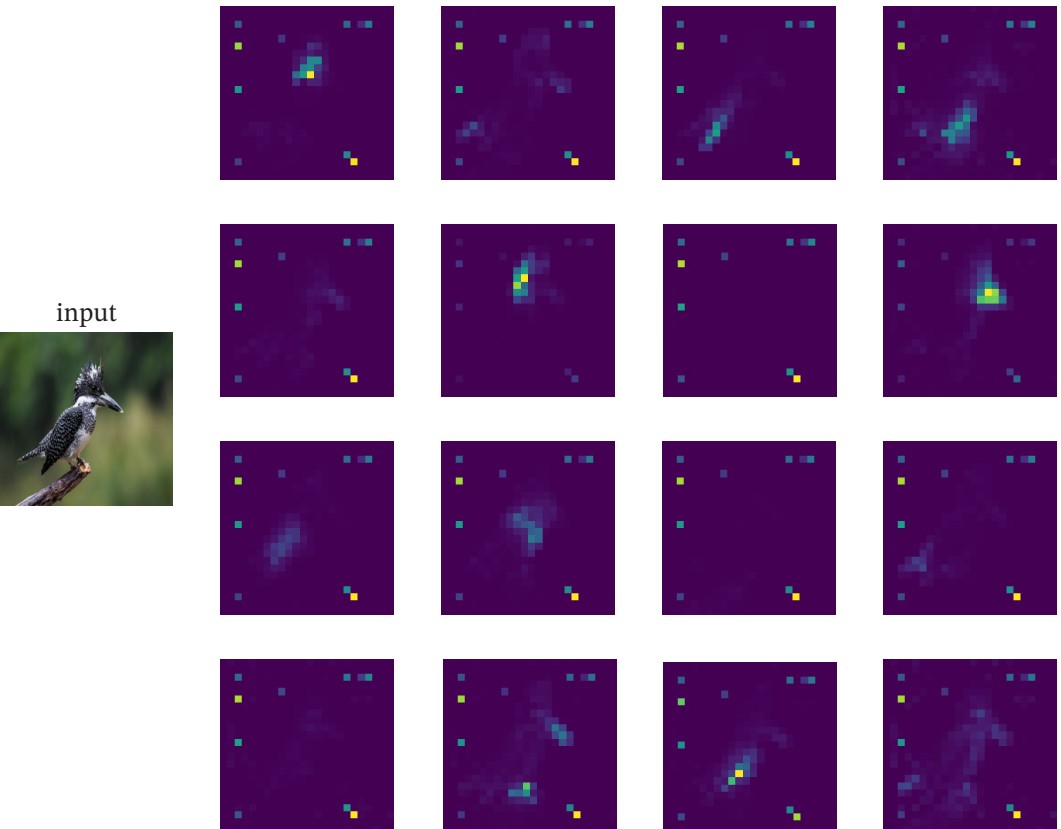

Figure 18: Attention maps of the `[CLS]` token to the patch tokens, shown here separately per attention head. We produce these maps with a DINOv2-L model trained without registers.

| dataset token | Airc. | CF10 | CF100 | CUB | Cal101 | Cars | DTD |
|---|---|---|---|---|---|---|---|
| normal | $17.1_{\pm0.5}$ | $97.1_{\pm0.1}$ | $81.3_{\pm0.3}$ | $18.6_{\pm0.6}$ | $73.2_{\pm1.3}$ | $10.8_{\pm0.3}$ | $63.1_{\pm0.8}$ |
| outlier | $79.1_{\pm0.5}$ | $99.3_{\pm0.0}$ | $93.7_{\pm0.3}$ | $84.9_{\pm2.1}$ | $97.6_{\pm0.7}$ | $85.2_{\pm0.9}$ | $84.9_{\pm0.9}$ |
| `[CLS]` | 87.3 | 99.4 | 94.5 | 91.3 | 96.9 | 91.5 | 85.2 |

| dataset token | Flow. | Food | IN1k | P205 | Pets | SUN | VOC |
|---|---|---|---|---|---|---|---|
| normal | $59.5_{\pm1.2}$ | $74.2_{\pm0.3}$ | $65.8_{\pm0.1}$ | $53.1_{\pm0.3}$ | $47.8_{\pm0.5}$ | $37.7_{\pm0.3}$ | $70.8_{\pm0.5}$ |
| outlier | $99.6_{\pm0.0}$ | $93.5_{\pm0.2}$ | $69.0_{\pm0.7}$ | $55.1_{\pm1.0}$ | $94.1_{\pm0.2}$ | $78.5_{\pm0.2}$ | $89.7_{\pm0.1}$ |
| `[CLS]` | 99.7 | 94.7 | 86.0 | 66.4 | 96.9 | 78.6 | 89.1 |

Table 6: Image classification via linear probing on normal and outlier patch tokens. As we select the patch tokens randomly among the set of eligible tokens, this adds a source of variability. We report the standard deviation of this variability in grey along with the scores. This table is a detailed view of table 1.

# H    QUALITATIVE RESULTS

We trained three popular models: DeiT-III, OpenCLIP, DINOv2 with and without the introduction of register tokens. We observe in Fig. 19 the attention maps in the last layer of the Vision Transformer, for all three cases. We see that our approach provides much cleaner attention maps, with considerably fewer artifacts, explaining the improvement on the downstream object discovery task mentioned in Sec. 3.3. The feature maps are also visibly improved, as shown in Fig. 20. Finally,

we also show the norm of the patch tokens in Fig. 21, and confirm that in all three models, artifact patches correspond to norm outliers.

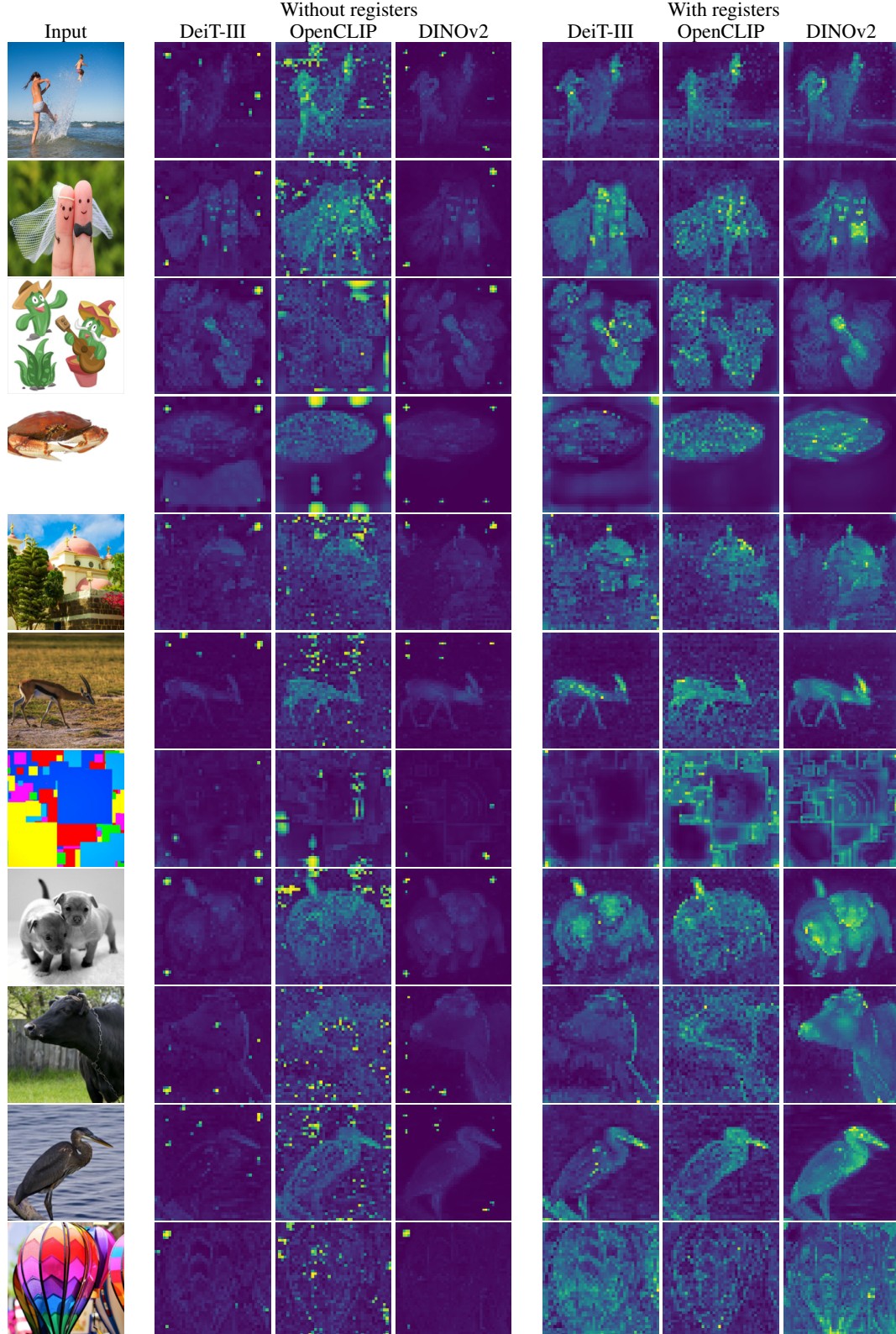

Figure 19: Attention maps of models trained without and with registers on various images.

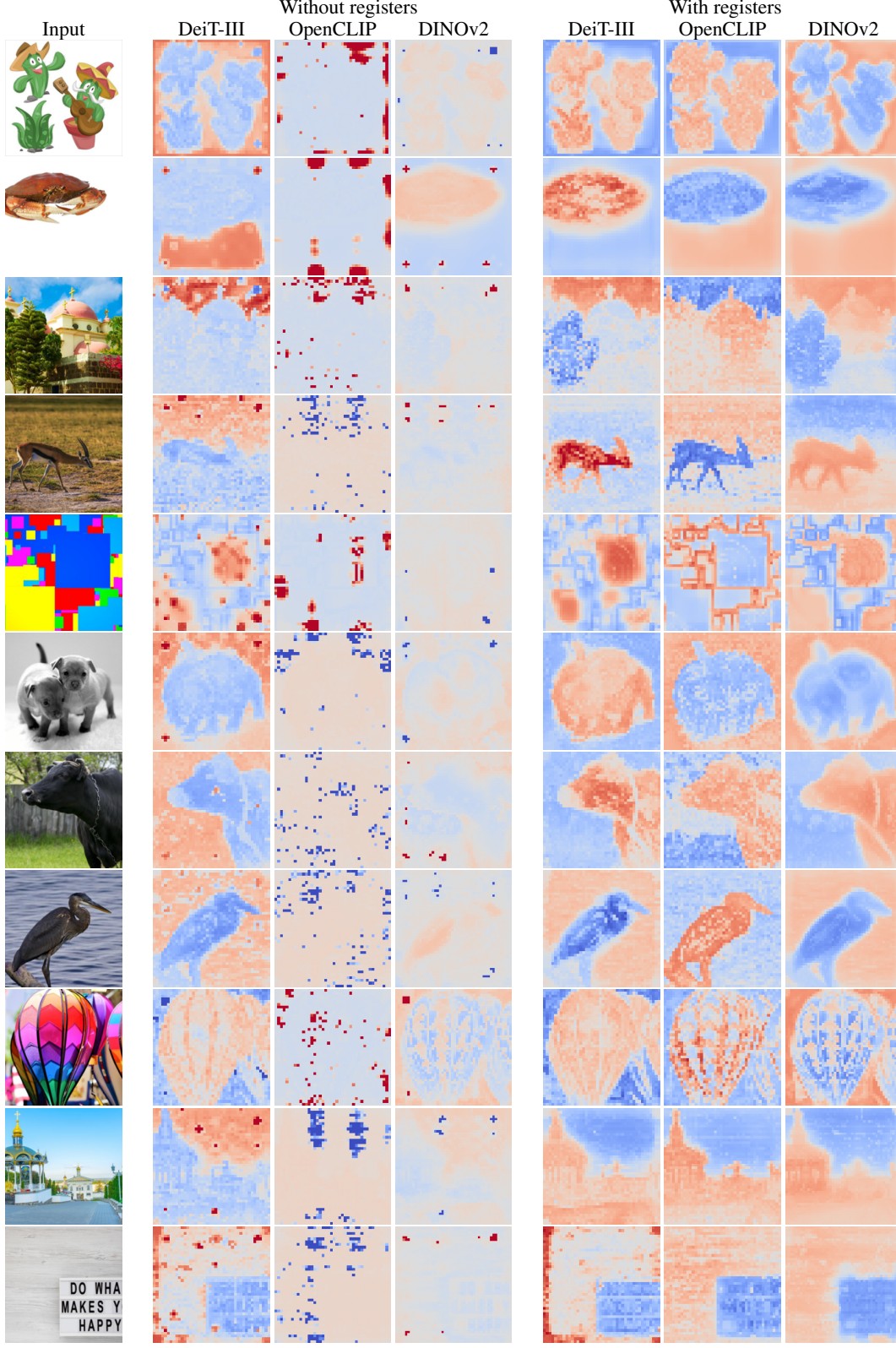

Figure 20: First principal component of the feature maps output by models trained without and with registers on various images. The components are whitened and the colormap covers the range $[-3\sigma, +3\sigma]$.

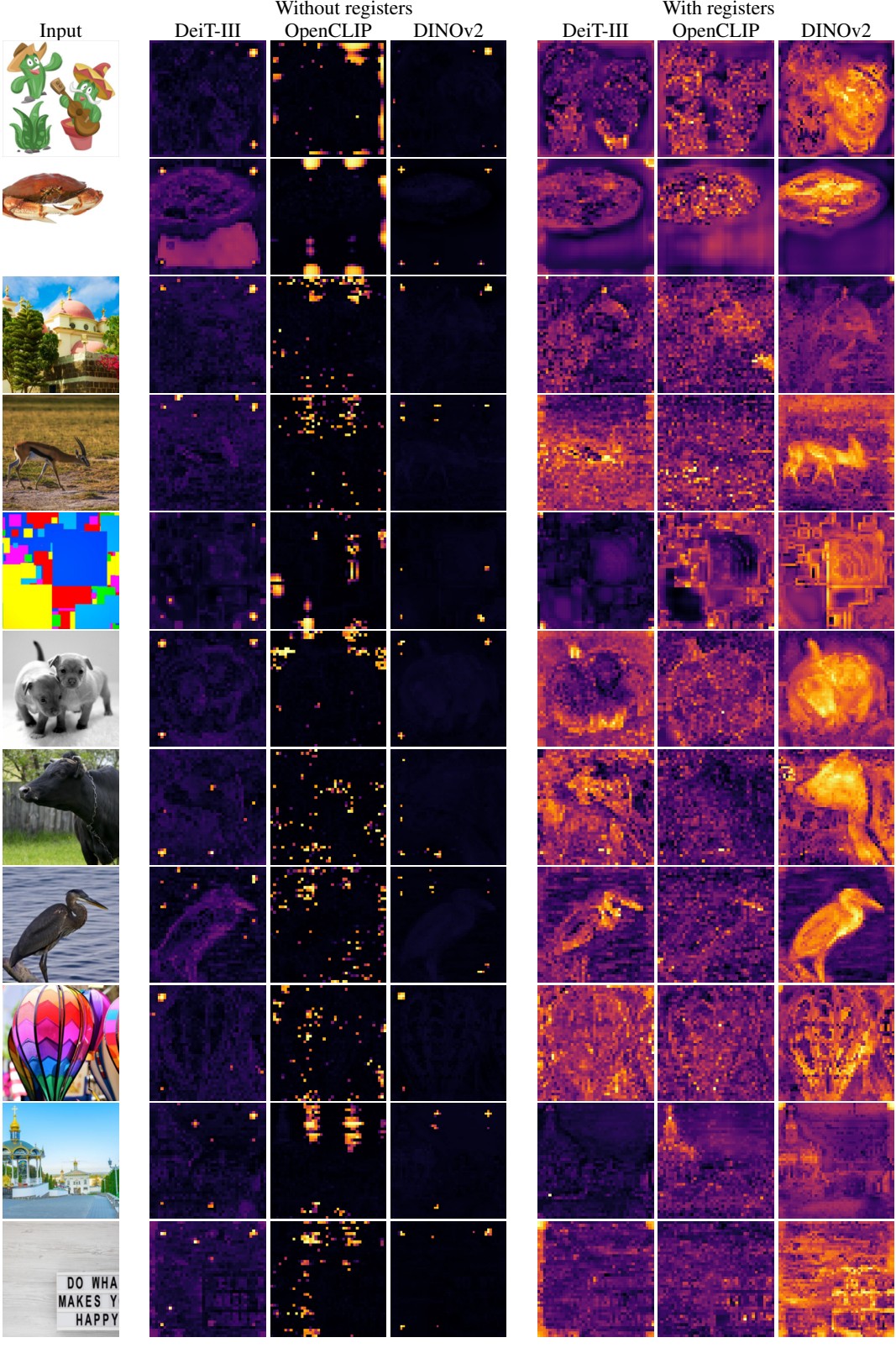

Figure 21: Maps of token norms for models trained without and with registers on various images. The norm outliers are very visible for models trained without registers.

