# OpenReview forum: "Vision Transformers Need Registers"
_ICLR.cc/2024/Conference — ICLR 2024 oral_

### Official Review · Reviewer_mTMB · 2023-10-26

**Soundness:** 3 good
**Presentation:** 4 excellent
**Contribution:** 4 excellent
**Rating:** 8
**Confidence:** 4

**Summary:**

This paper identifies an interesting phenomena in large scale transformers where some redundant tokens are repurposed for internal computation. The paper shows how the feature normal can be used to identify such tokens and how such tokens appear to capture global, rather than local, information compared to other tokens. Furthermore, such tokens make the attention maps less interpretable. The paper proposes to augment ViTs with register tokens which similar to CLS tokens are separate from the image patch tokens, but are not used used directly in any loss computations unlike CLS tokens. The proposed augmentation removes the normal outliers, results in small improvements on standard evaluation tasks, while improving the unsupervised object discovery performance of most methods.

**Update (11/20):** I have updated my scores after reading the points made by other reviewers. Specifically, the point regarding the impact of the tokens on dense vs. image-level tasks, specifically weakness 3 raised by reviewer KSLu. It would be great if the authors could engage with the concerns raised by the reviewers.

**Strengths:**

- The analysis of the outlier tokens is very nice and thorough. I found the graphs and explanations very insightful, and the experiments very comprehensive (especially the experiments in Tab 1 and Fig 5).
- The proposed inclusion of a register token is very simple and elegant and provides more interpretable attention masks.
- I appreciated the limitations statement at the end of Sec 2.2.
- The paper was very easy to follow and the visualizations were helpful to provide the reader intuition for what is going on.

**Weaknesses:**

- While the paper did a great job at analyzing the behavior of outlier tokens in previous models in Sec 2.1, the paper does not have experiments showing that such behavior is eliminated by adding the register tokens. It would have been interesting to see if the behaviors ascribed to normal/outlier tokens in Fig 5 and table 1 are now transferred to image/register tokens in the proposed model.
- The discussion around the performance of models on unsupervised object discovery is fairly limited and does not match the resuls.
    - The paper is strongly motivated by the difference in attention maps compared to DINO and the limited performance of DINOv2 on LOST. While the gains of DINOv2+reg are impressive, it is still very surprising that it doesn't match DINO. It would be great if there was more discussions or some qualitative examples of that to explain why.
    - The paper states that "for all models on all datasets, adding registers for training improves the unsupervised object discovery performance." However, the results indicate that registers harm the performance on OpenCLIP.

**Questions:**

- Do the registers inherit the behavior exhbited by the outlier tokens? Specifically, are they good predictors of global image information as shown in Table 1?
- Could you clarify on the discrepancy in OpenCLIP performance in Table 3 vs Sec 3.3? I wasn't sure if it's a missed negative result or a typo in the table.
- Do the image tokens revert back to being more local in nature with the addition of tokens? How do they perform on the tasks exhibited in Table 1 and Figure 5?
- How does the norm of the CLS token compare to the outliers before and after the addition of register tokens? It would be interesting to see if it matches the outlier tokens across conditions as shown in Figure 4, although simply reporting it on the final model would provide some insight into the internal mechanisms of ViTs.
- Could you please comment on why you think LOST with DINO still performs better that with DINOv2? Is it the data (ImageNet vs LVD), newer training objective, some other factor?
- It seems suprising that DINOv2 exhibits this behavior while using a dense mask-image-modeling objective. I was curious if you had any thoughts on why the masked image objective did not discourage such behavior despite requiring the patch features to retain the information that you show is lost in Fig 5b
- I found the first line in page 9 a bit confusing. As I understand it, Torralba and Efros (2011) were arguing that datasets themselves were biased, not that the specific labels were. Such concerns still apply whether or not the data is labels or the training paradigm used for training as the bias arises from the data source and sampling process. While the samping process would be affected by the target labels, one can still get a biased dataset based solely on the data source (eg, instagram vs. inaturalist). Could you please elaborate on your statement and what you meant?
- (Suggestion) The paper suggests that this outlier token is exhibited by larger models in Fig 4/Sec 2.2, yet the base model of CLIP/DEIT is used in Sec 3.1. While Fig 7 still shows that the base models of those models exhibit outlier tokens, it would be nice for the authors to add some commentary on this at the end of sec 2.2.

---

> ### Author Response · Authors · 2023-11-22
> **Answers to the questions raised in review**
>
> We thank reviewer mTMB for their remarks and their thoughtful review. We answer the questions below:
>
> > Do the registers inherit the behavior exhibited by the outlier tokens? Specifically, are they good predictors of global image information as shown in Table 1?
>
> It appears that registers inherit the behavior of outliers. We detail this analysis in Appendix D in the paper and thank the reviewer for this question. This additional experiment fills a gap in our analysis.
>
>
> > Could you clarify on the discrepancy in OpenCLIP performance in Table 3 vs Sec 3.3? I wasn't sure if it's a missed negative result or a typo in the table.
>
> Reviewer eoPK brought up the same point. We apologize for the mistake in the text due to an oversight on our part; the claim that registers improve object discovery in all models is not supported by the numbers presented, as there is a slight loss in performance for the OpenCLIP model. We corrected this and improved the main text accordingly in Section 3.3. In order to complement this observation, we provide a more thorough analysis in Appendix C.
>
>
> > Do the image tokens revert back to being more local in nature with the addition of tokens? How do they perform on the tasks exhibited in Table 1 and Figure 5?
>
> We performed the corresponding measurements in Table 5 (Appendix D2) and show that the image tokens (in a model trained with registers) match the performance of non-outlier tokens (in a model trained without registers).
>
>
> > How does the norm of the CLS token compare to the outliers before and after the addition of register tokens? It would be interesting to see if it matches the outlier tokens across conditions as shown in Figure 4, although simply reporting it on the final model would provide some insight into the internal mechanisms of ViTs.
>
> We conducted an additional experiment to study this matter. Appendix D.1 shows a plot of the norms of all the tokens output by the vision transformer, split by token type. We compute these norms on a random sample of images taken from ImageNet-22k. We observe that with and without registers, the norm of the CLS token is consistently low, with no outliers. Interestingly, the high-norm outliers observed in the patch tokens now appear in the register tokens.
>
>
> > Could you please comment on why you think LOST with DINO still performs better that with DINOv2? Is it the data (ImageNet vs LVD), newer training objective, some other factor?
>
> We do not have a decisive answer to that question, and can only formulate initial hypotheses :
> - First, the LOST method was specifically designed in conjunction with the DINO models. The heuristics involved in this work were tuned specifically for the attention maps of DINO models. Our intuition is that some amount of performance was lost to this procedure: we have not spent as much effort tweaking this method - it is not part of our core contribution.
> - Second, DINOv2 tends to discern regions more finely, distinguishing object parts on top of full objects. Our intuition is that DINO was trained in a more object-centric setup (no masked image modeling, training on ImageNet-1k), which fits the object discovery setup better.
>
>
>
> > It seems suprising that DINOv2 exhibits this behavior while using a dense mask-image-modeling objective. I was curious if you had any thoughts on why the masked image objective did not discourage such behavior despite requiring the patch features to retain the information that you show is lost in Fig 5b
>
> This is a very good question. In DINOv2, the masked image modeling loss is applied on [MASK] tokens with a position embedding, and not on visible image tokens. In informal experiments, we have observed that the outliers appear only on visible tokens and never on [MASK] tokens. The visible patch tokens themselves are not used for computing any loss, meaning their high norm is not discouraged by the training objective. However, the fact that outliers do not appear on [MASK] tokens suggests, in line with the remark of the reviewer, that receiving a loss signal discourages outliers.

---

> > ### Author Response · Authors · 2023-11-22
> >
> > > I found the first line in page 9 a bit confusing. As I understand it, Torralba and Efros (2011) were arguing that datasets themselves were biased, not that the specific labels were. Such concerns still apply whether or not the data is labels or the training paradigm used for training as the bias arises from the data source and sampling process. While the samping process would be affected by the target labels, one can still get a biased dataset based solely on the data source (eg, instagram vs. inaturalist). Could you please elaborate on your statement and what you meant?
> >
> > Thanks for pointing that out; you are right that the Torralba & Efros paper focuses on the datasets themselves, and the citation was mainly to acknowledge that they have popularized the term “dataset bias”; however your question is sound and we would like to further clarify what we meant in the following way:
> >
> > We believe datasets are biased in at least two ways: how the labels are determined and how the images are sampled. SSL can negate some of the bias related to labels. For example, one can imagine a dataset of dog images, labeled as male/female. The (image, labels) pairs would, in that case, entirely ignore the breeds of the dogs. In contrast, SSL would attempt to cluster samples from a given breed together.
> >
> > Additionally, as SSL does not rely on labels, multiple datasets can be easily concatenated together for training jointly, as the labels do not need to be compatible. This would, in particular, resonate with the “Negative Set Bias” paragraph at the end of the Torralba & Efros paper, where the authors recommend using negatives from other datasets in order to model the rest of the visual world better and therefore reduce image selection biases.
> >
> > If possible, we would like to get feedback from Reviewer mTMB. If our analysis is reasonable, we will update the text with a reformulation of the ideas presented above. Otherwise, we will clarify to separate dataset biases from labeling biases and avoid confusion related to citing that paper.
> >
> >
> >
> > > (Suggestion) The paper suggests that this outlier token is exhibited by larger models in Fig 4/Sec 2.2, yet the base model of CLIP/DEIT is used in Sec 3.1. While Fig 7 still shows that the base models of those models exhibit outlier tokens, it would be nice for the authors to add some commentary on this at the end of sec 2.2.
> >
> >
> > It seems that the conditions for the apparition of these outliers are multiple. Both larger models and models trained for longer seem more vulnerable. However, OpenCLIP and DeiT-III indeed show outliers at sizes smaller than DINOv2, indicating that the pretraining paradigm does also play a significant role. We agree that this may have been unclear, and have updated section 2.2 with a discussion on this part.

---

> ### Comment · Reviewer_mTMB · 2023-11-22
>
> Thank you for responding to my concerns and for your engagement. I appreciated the additional analysis (especially Appendix D) and elaboration of the phenomena. I really appreciate the author's clear delineation between explanations supported by evidence vs. points they hypothesize or speculate about which allows the reader to easily understand the statements and contextualize the confidence behind them appropriately. Overall, my concerns have been addressed, and the discussion below is just engagement regarding the paper and is more subjective in nature than the review.
>
> **Connection to dataset bias**:
> I agree with the separation of two forms of bias. I will note that for curated datasets such as ImageNet and LVD, those biases are very closely intertwined due to the sampling being the direct result of either querying with keywords or clustered representations that depend on a specific class set. Although, those two factors likely affect LVD less due to the use of image features as queries.
>
> While the Torralba and Efros suggested using negatives, the results they report are quite mixed where it hurts performance on same datasets and benefits on others. The negative impact is attributed to negatives being correlated with the class, while the improvement on ImageNet is attributed to data variability. So it is unclear how applicable that statement is to the methods. At a high level, I agree with your analysis of two kinds of bias and that SSL is less susceptible to it, although curation might be a culprit here (as I discuss next).
>
> It seems to me that while SSL does not require labels, they might require some amount of data curation. This is speculation on my side as I do not have concrete evidence to support this beyond some smaller scale experiments where I found that they perform poorly for less curated datasets. Some work in the literature does support this; eg, Assran et al [1] very nicely discusses the impact of the uniformity in dataset training and suggests that this is benefiting SSL methods. While they show results on subsets of the data sampled in various ways, the underlying data they are sampling from is still relatively curated. Furthermore, it remains unclear how DINO or DINOv2 would perform if trained on very large scale datasets without curation like LAION. Of course, I think this is an open question and lies well beyond the scope of your work, but one that is worth considering when contextualizing some of the results with respect to biases in the data or trying to understand the interplay between different learning signals and different types of data.
>
> One final thing that I thought might be interest is the impact of the choice of layer for conducting this analysis, especially when thinking about the CLIP numbers. Walmer et al [1] similarly analyze the features learned by transformers and find that in some cases, representations of intermediate layers have more information regarding specific objectives such as clustering for parts vs. objects. I was curious if some of those findings would impact the analysis does in Appendix C as well as your second hypothesis regarding the difference between DINO and DINOv2's performance on LOST being attributed to the granularity of objects/parts being represented. I think that your point about MIM encouraging more part representations is likely the correct explanation, but it is also possible that it simply shifts where different aspects of the image are represented in the network.
>
>
> References:
>   1. The hidden uniform cluster prior in self-supervised learning  https://openreview.net/pdf?id=04K3PMtMckp
>   2. Teaching Matters: Investigating the Role of Supervision in Vision Transformers https://arxiv.org/abs/2212.03862

---

### Official Review · Reviewer_KSLu · 2023-10-31

**Soundness:** 3 good
**Presentation:** 4 excellent
**Contribution:** 3 good
**Rating:** 8
**Confidence:** 4

**Summary:**

This paper identifies and characterizes artifacts in the feature maps of vision transformer (ViT) models trained with supervision or self-supervision. In particular, the authors observe high-norm "outlier" tokens with high redundancy in the output features of several ViT models, and show that they hold less local patch information but more global image information compared to normal tokens. This suggests the model is repurposing redundant patches to store global information. They propose appending dedicated "register" tokens to the input sequence, which removes the artifacts and improves performance on downstream dense prediction tasks.

**Strengths:**

1. The investigation is quite original; the use of memory/registers in transformers is not necessarily a new idea, but motivating them through removing redundancy and reducing attention artifacts is both novel and interesting.

2. Experiments and analysis are mostly convincing (see questions below).

3. I enjoyed the narrative exposition: the problem setting is clear, the motivation for registers is clear, and their utility is well-demonstrated via experiments.

**Weaknesses:**

1. While adding additional token (registers) seems like a simple and efficacious approach, I'm wondering if it's the only possible solution for reducing patch level redundancy. Did the authors observe similar effects across other self-supervised models, like MAE, where nominally the patch-level reconstruction should also alleviate representational redundancy?

2. In demonstrating that the artifacts hold global information, the authors "choose a single token at random, either high-norm or normal," and then "train a logistic regression classifier to predict the image class from this representation, and measure the accuracy." Why choose this token at random? Why not use all the high-norm and normal tokens, or some projected and pooled version over all of them in order to regress to the class? In experiments we have conducted, this almost always outperforms using single tokens (cls or otherwise), and it may be the case that the conclusion that the high-norm tokens outperform the "normal" tokens is not so clear when this is done.

2. There seems to be a conflict between the fact that "high-norm tokens appear on patches that are very similar to their neighbors,"  "often appear[ing] in uniform, background areas," and the fact that performance on ImageNet classification improves almost monotonically with more registers, but not dense tasks like segmentation or depth estimation. In particular, I would expect that if reappropriating the "redundant" local patches helps on object-centric classification, it should help much more substantially on tasks that are even more reliant on good (non-redundant?) local information (i.e. segmentation or depth estimation).  Can the authors comment on this?

**Questions:**

See weaknesses above.

---

> ### Author Response · Authors · 2023-11-22
> **Answers to the questions raised in review**
>
> We thank reviewer KSLu for their remarks and their thoughtful review. We answer the questions below:
>
>
> > 1. While adding additional token (registers) seems like a simple and efficacious approach, I'm wondering if it's the only possible solution for reducing patch level redundancy. Did the authors observe similar effects across other self-supervised models, like MAE, where nominally the patch-level reconstruction should also alleviate representational redundancy?
>
> We agree there probably exist other solutions that could work. On MAE, our experiments seem to show that it does not exhibit the same "outlier patches" as the other models. We believe it may be linked to the fact that MAE was trained only with a local loss, there might be no need (or less need) to aggregate global information anywhere, and thus these outliers may not be needed. However, we also believe that relying only on a local loss is the reason leading to a poor performance for representation learning: MAE-ViT-Large only reaches 75% classification accuracy with linear probing, which is way below the other SSL methods.
> We expand on this in Appendix E.
>
>
> > 2. In demonstrating that the artifacts hold global information, the authors "choose a single token at random, either high-norm or normal," and then "train a logistic regression classifier to predict the image class from this representation, and measure the accuracy." Why choose this token at random? Why not use all the high-norm and normal tokens, or some projected and pooled version over all of them in order to regress to the class? In experiments we have conducted, this almost always outperforms using single tokens (cls or otherwise), and it may be the case that the conclusion that the high-norm tokens outperform the "normal" tokens is not so clear when this is done.
>
> The goal of this experiment is to compare the global information contained in the different kinds of patches (individually) and assess whether outlier tokens are closer to the CLS token, which holds global information, or closer to non-outlier patch tokens, that hold local information.
>
> In order to perform token-to-token comparison, we estimate the average performance for individual patches over the dataset, and randomly choosing a token allows this estimation; in order to confirm that this approach is sound, we also provide standard deviation numbers in the manuscript (see updated Table 6 in Appendix G) obtained across runs.
>
> While we agree that the performance of average-pooled patches is expected to be much stronger than individual patches, this appears misaligned with our goal of characterizing the outlier tokens in contrast to the other token types individually.
>
>
>
> > 3. There seems to be a conflict between the fact that "high-norm tokens appear on patches that are very similar to their neighbors," "often appear[ing] in uniform, background areas," and the fact that performance on ImageNet classification improves almost monotonically with more registers, but not dense tasks like segmentation or depth estimation. In particular, I would expect that if reappropriating the "redundant" local patches helps on object-centric classification, it should help much more substantially on tasks that are even more reliant on good (non-redundant?) local information (i.e. segmentation or depth estimation). Can the authors comment on this?
>
> Thanks for pointing out this apparent conflict. Adding register tokens has two effects :
> - First, adding registers removes the need for high-norm artifacts in the feature maps. This effect is visible with one register both qualitatively and quantitatively (see Fig. 8). With n=1, artifacts disappear from the attention maps, ImageNet classification accuracy is unchanged (+0.05), while segmentation and depth prediction are significantly improved (+0.6 mIoU and -0.1 RMSE).
> - Second, when adding more registers, another behavior emerges. Segmentation and depth prediction performance does not improve much (because the feature maps are now already clean), but classification performance improves further. We agree with the reviewer that this is surprising; we do not have a clear intuition yet of why additional registers improve classification performance and hope this can be answered in future research.

---

### Official Review · Reviewer_GGy7 · 2023-11-01

**Soundness:** 3 good
**Presentation:** 4 excellent
**Contribution:** 4 excellent
**Rating:** 8
**Confidence:** 4

**Summary:**

The paper identifies the problem of artifactual areas of feature maps in vision transformers. On further analysis, these artifacts correspond to high norm tokens in the ViT coming from background regions in the image, tend to hold more global information and lack spatial information. This leads to the conclusion that tokens from these low-information regions are being repurposed by the model to hold global information for internal computations. Moreover, this issue afflicts most ViTs with DINO v1 being the exception. This problem was previously discussed in Memory transformer (Burtsev et al.) in the context of NLP datasets. Following Memory transformer's recommendation, the paper adds new tokens called registers at the to remediate this issue and show that the register-trained ViTs have better spatial feature maps, thus better downstream performance for object discovery tasks. Various ablations and experiments are also shown to shed light on the behavior of registers and why this problem occurs with DINOv2 models in the first place.

**Strengths:**

1. The paper identifies an important problem of heatmaps lacking spatial resolution and accuracy in DINOv2 and other ViTs which leads to suboptimal downstream performance on object discovery and localization tasks. The fact that this is not a problem for DINOv1 is pretty surprising and the experiments done on the changes in token norm across model size will be a useful start to understand this better. The discovery of these high-norm tokens, experiments using the linear models and classifiers to characterize what these tokens contain, and then forming the hypothesis of how these tokens are getting repurposed for holding global information, all presents a coherent story of these misused outlier tokens.
2. The solution of adding new tokens (memory or registers) is not a new one, but is shown to be very effective in removing these tokens with high norms as well as bringing back spatial interpretability into the feature maps. Furthermore, the downstream performance on image-level tasks stain consistent with ViTs without registers while the performance on object discovery tasks goes up in DINOv2 and DeiT-III after addition of registers. These results indicate that adding these new tokens does resolve the spatial issue with these ViTs. A huge plus of this approach is the simplicity of it.

**Weaknesses:**

1. The removal of these artifacts does come at the cost of new tokens, hence additional compute. The paper reports a 2-6% increase when adding 4-16 new register tokens.
2. One very interesting observation was how the different register tokens end up focussing on the different areas of interest on the object. If there are spatially discrete areas of focus for the registers, does this undermine the argument that we need them for storing global information which was earlier being done using redundant patches?
3. Not a weakness, but would be nice to see some norm-related metrics and/or visualizations for the outlier tokens across different heads. Do all the heads from these tokens end up getting these tokens repurposed? What does that variance across heads look like?

**Questions:**

1. I'm curious if other simple solutions like penalizing these high norms could work as well and if yes, would they be preferable? Would love to hear from the authors on other ways to address this issue.
2. Any reason why adding more registers hurts performance for NYU depth dataset?
3. What happens to downstream performance when registers are added to DINO models which do not seem to need it? Does the nature of what registers learn differ from the case of DINOv2?

---

> ### Author Response · Authors · 2023-11-22
> **Answers to the questions raised in review**
>
> We thank reviewer GGy7 for their remarks and their thoughtful review. We answer the questions below:
>
> Although this was not noted as a weakness, we would like to add a precision about the relationship with Memory Transformers (Burtsev et al.). We included this reference as it is, to our knowledge, the only previous case of adding similar additional tokens to the input sequence in transformers. However, this was only done in the goal of increasing the scores for NLP tasks. The analysis of artifacts in the features, and the idea of adding new tokens to fix these artifacts, is specific to our analysis. We hope this clarifies the relationship with the previous literature.
>
> > One very interesting observation was how the different register tokens end up focussing on the different areas of interest on the object. If there are spatially discrete areas of focus for the registers, does this undermine the argument that we need them for storing global information which was earlier being done using redundant patches?
>
> We added visualizations describing the positional focus of the different registers, by showing averaged attention maps over a dataset, in Appendix D3. It appears that the areas of focus for the registers have a larger extent on average and cover wider areas, similar to the CLS token, and different of the patch tokens that are much more local in nature. We want to point out that in the example in Figure 9, the registers attend to all objects present in the image (hence global), but with a slightly stronger focus on individual objects depending on the register observed. Therefore, the focus is not discrete and we think it does not undermine the argument around storing global information.
>
> > Not a weakness, but would be nice to see some norm-related metrics and/or visualizations for the outlier tokens across different heads. Do all the heads from these tokens end up getting these tokens repurposed? What does that variance across heads look like?
>
> We added a visualization per attention head for the outliers, in Appendix G. The analysis shows that most heads are similarly affected by the presence of outliers, with a few heads focusing a bit more on the objects.
>
> > I'm curious if other simple solutions like penalizing these high norms could work as well and if yes, would they be preferable? Would love to hear from the authors on other ways to address this issue.
>
> It is definitely possible that other valid solutions may exist. For the context of this paper, we preferred focusing on what we felt was the most natural fix, but penalizing the high norms of the patch tokens may be indeed another possibility. To study this hypothesis, we launched a few pretraining runs of DINOv2 ViT-L with a penalization on the L2 norm with various hyperparameters. The result of these runs is not available yet, but we believe that it might be treating a symptom rather than the cause: if the model still needs some place to store global information, it might still do it while keeping the norms low. In contrast to our approach, regularization of the norms induces more hyperparameters that can be difficult to tune, and no opportunities for exploiting outputs of the model.
>
> > Any reason why adding more registers hurts performance for NYU depth dataset?
>
> When going from 8 to 16 registers, the sequence length is increased by a non-negligible amount, and this can lead to a different optimal set of hyperparameters for training. Since the Fig.8 experiment was conducted with constant hyperparameters, the optimisation may be a culprit for the slight increase in rmse when using 16 registers. The goal of the figure was to point out that there was a clear step effect when going from 0 to 1 register (matching the qualitative effect of removing artifacts), and only smaller effects when adding more registers.
>
> Alternatively, the small difference in results (0.03 RMSE for 8->16 regs, compared to 0.1 RMSE for 0->1 reg) could be simply noise.
>
>
> > What happens to downstream performance when registers are added to DINO models which do not seem to need it? Does the nature of what registers learn differ from the case of DINOv2?
>
> This is an interesting idea, and we are not sure of the answer. We do not expect the patch tokens to be improved (as there are no outliers), but the classification performance may increase (as in fig. 8).
> In order to understand this better, we launched a new pretraining of DINOv2 ViT-B with 4 registers. Since ViT-B does not exhibit outliers (fig. 4.c), this should give us an empirical answer to this question, and provide additional insights for understanding registers better.

---

### Official Review · Reviewer_eoPK · 2023-11-09

**Soundness:** 3 good
**Presentation:** 3 good
**Contribution:** 3 good
**Rating:** 8
**Confidence:** 3

**Summary:**

The paper discusses the discovery of artifacts in the feature maps of Vision Transformer (ViT) networks, both supervised and self-supervised. These artifacts appear as high-norm tokens during inference, typically in less informative background areas of images, and are utilized for internal computations by the network. To address this, the authors introduce a novel and straightforward method involving the addition of extra tokens to the ViT's input sequence. This technique effectively resolves the artifact issue for both types of models. It not only sets new performance benchmarks for self-supervised visual models on dense prediction tasks but also enhances object discovery with larger models. Crucially, the approach results in smoother feature and attention maps that benefit subsequent visual processing tasks.

**Strengths:**

- The paper identifies an interesting phenomenon observed in the popular transformer models (DINO). By removing this artifact, the authors demonstrates the improved models have clear attention maps that could be used for downstream analysis such as object localization.
- The step-by-step investigation is solid and compelling.
- The method of providing a junkyard to remove the artifact is novel and effective.
- The experiments are convincing and comprehensive.

**Weaknesses:**

- The norm shows significant reduction for OpenCLIP in Figure 7, yet in Table 3, it doesn’t show significant improvement for object localization which is the main benefit of using register. Further explaination / exploration the reason behind it should be helpful for wide adoptation.
- Minor: it should be OpenCLIP instead of CLIP in Figure7.

**Questions:**

In table 3, OpenCLIP+reg is not better than OpenCLIP which is contrary to other two models that have significant improvement. Any further explaination could be helpful since it doesn’t support the claim that for all models, adding registers would improves the results.

---

> ### Author Response · Authors · 2023-11-22
> **Answers to the questions raised in review**
>
> We thank reviewer eoPK for their remarks and their thoughtful review. We answer the questions below:
>
> > In table 3, OpenCLIP+reg is not better than OpenCLIP which is contrary to other two models that have significant improvement. Any further explaination could be helpful since it doesn’t support the claim that for all models, adding registers would improves the results.
>
> We apologize for the mistake in the text due to an oversight on our part; the claim that registers improve object discovery in all models is not supported by the numbers presented, as there is a small loss in performance for the OpenCLIP model. We correct this and improve the main text accordingly in Section 3.3.
>
> > The norm shows significant reduction for OpenCLIP in Figure 7, yet in Table 3, it doesn’t show significant improvement for object localization which is the main benefit of using register. Further explaination / exploration the reason behind it should be helpful for wide adoptation.
>
> Regarding the incoherence between Table 3 and Fig. 7: we have also found this surprising and conducted additional experiments. In our evaluation on unsupervised object discovery, for each model, we select the best performing embedding (keys, queries, values). For CLIP, this turns out to be the values. In Fig. 14, we show the seed expansion score obtained in LOST using k, q or v. We clearly see that the artifacts are visible when using keys or queries, but not values. This is therefore coherent both with the quantitative results in Table 3, qualitative analysis in Fig. 13 and the observation raised by reviewer eoPK about Fig. 7. We added a discussion on that matter in Appendix C. We thank the reviewer for raising this point as this analysis improves the coherence of the presentation.
>
> > Minor: it should be OpenCLIP instead of CLIP in Figure7.
>
> We replaced CLIP by OpenCLIP in the labels in Fig. 7.

---

### Author Response · Authors · 2023-11-22
**New revision**

We want to thank the reviewers for their thoughtful comments and questions, and we feel the subsequent additions make the paper much stronger and sound. We have updated the manuscript with a new revision, modifying the main text and adding appendices with more experimental evidence to support our answers in the discussions with reviewers.

---

### Public Comment · ~Swarnim_Maheshwari1 · 2024-03-23
**What is the difference between learnable prompt and register tokens?**

A learnable prompt is a set of tokens that are prepended or appended to the input prompt. They are initialized randomly and thus both learnable prompts and register don't provide additional information to the model.

Could the authors please clarify the difference between the learnable prompt and register tokens as used in this work?

---

### Meta-Review · Area_Chair_rocL · 2023-12-11

**Metareview:**

This paper introduces a novel solution to address artifacts in the feature maps of ViT models, characterized by high-norm tokens associated with less informative background regions. The authors propose adding register tokens to the ViT's input sequence, effectively mitigating the artifact issue and enhancing the model's performance on various tasks.

The paper received very positive ratings (8, 8, 8, 8), and all the reviewers acknowledged the technical contribution presented in the paper. At the same time, reviewers raised several concerns, including discrepancies in performance for OpenCLIP, the relationship between dataset biases and SSL models, the choice of optimization formulation, and the impact of gradient clipping. The authors properly addressed these concerns and conducted additional experiments in their rebuttal.

In conclusion, all reviewers agreed that this paper is strong and recommended it for acceptance. Congratulations to the authors on their excellent work!

**Justification For Why Not Higher Score:**

N/A

**Justification For Why Not Lower Score:**

As mentioned, all reviewers agreed that this paper is strong and recommended it for acceptance.

---

### Decision · Program_Chairs · 2024-01-16

Accept (oral)